# Short- and Long-Term Endothelial Inflammation Have Distinct Effects and Overlap with Signatures of Cellular Senescence

**DOI:** 10.3390/cells14110806

**Published:** 2025-05-30

**Authors:** Barbora Belakova, José Basílio, Manuel Campos-Medina, Anna F. P. Sommer, Adrianna Gielecińska, Ulrike Resch, Johannes A. Schmid

**Affiliations:** 1Institute of Vascular Biology and Thrombosis Research, Center for Physiology and Pharmacology, Medical University of Vienna, 1090 Vienna, Austria; barbora.belakova@meduniwien.ac.at (B.B.); ruiz41@hotmail.com (M.C.-M.); ulrike.resch@meduniwien.ac.at (U.R.); 2Institute of Pathophysiology and Allergy Research, Medical University of Vienna, 1090 Vienna, Austria; jose.basilio@meduniwien.ac.at; 3Department of Molecular Biotechnology and Genetics, Faculty of Biology and Environmental Protection, University of Lodz, 90-237 Lodz, Poland; adrianna.gielecinska@edu.uni.lodz.pl; 4Doctoral School of Exact and Natural Sciences, University of Lodz, 90-237 Lodz, Poland

**Keywords:** senescence, acute inflammation, chronic inflammation, transcriptomics, gene set enrichment analysis, molecular signatures, proliferation, mesenchymal transition, wound healing

## Abstract

This study investigates the interplay between cellular senescence and inflammation in human umbilical vein endothelial cells (HUVECs). We employed RNA sequencing to analyze gene expression changes in HUVECs subjected to replicative- or radiation-stress-induced senescence, and we compared these profiles with those of cells under acute or chronic TNFα-mediated inflammation. Our findings reveal that both senescence types exhibited significant upregulation of genes associated with epithelial- (or endothelial) mesenchymal transition (EMT) and inflammatory pathways, indicating a shared molecular response. Notably, chronic inflammation led to a pronounced EMT signature, while acute inflammation primarily activated classical inflammatory responses. Experimental validation confirmed reduced proliferation and increased secretion of pro-inflammatory cytokines (IL-6 and IL-8) in senescent and chronically inflamed cells and substantiated the upregulation of EMT marker genes. Additionally, we observed impaired wound healing capacity in senescent and chronically inflamed cells, highlighting the functional consequences of these cellular states. Our study underscores the critical role of inflammation in exacerbating senescence-related changes, contributing to the understanding of age-related cardiovascular pathologies. These insights may inform future therapeutic strategies aimed at mitigating the effects of aging and inflammation on endothelial function and cardiovascular health.

## 1. Introduction

The prevalence of cardiovascular disease in the world is currently increasing steadily with age and is the leading cause of death worldwide [1]. Aging is closely associated with the phenomenon of cellular senescence, a state in which cells dramatically alter their metabolic, morphological, and proliferative properties [2]. The concept of cellular senescence and the terminology were coined in the last century by Leonard Hayflick and Paul Moorhead in their pioneering work [3] where they demonstrated, contrary to the prevailing belief at that time, that primary cells would stop dividing after a finite number of population doublings. These initial observations led to the theory of aging according to which senescent cells accumulate with age in vivo or during prolonged culture in vitro, primarily due to telomere shortening. In addition, cellular senescence can be prematurely induced in a replication-independent manner, mainly in response to various cellular and/or DNA-damaging agents, including reactive oxygen species [4], anticancer drugs [5], inflammatory cytokines [6], mitochondrial dysfunction [7], high glucose concentrations [8], UV light [9], or ionizing radiation [10]. The main characteristic of senescent cells regardless of the inducing factor is the irreversible loss of proliferative potential, which in contrast to quiescent cells persists even in the presence of growth factors. This state is mainly maintained by the upregulation of cell cycle inhibitors [11]. In addition to cell cycle arrest, senescent cells acquire a pro-inflammatory “senescence-associated secretory phenotype” (SASP) which implies that cells are in a state of chronic low-grade inflammation. Core SASP factors include key players of NF-κB signaling pathways such as interleukin 6 or 8 (IL-6 or IL-8, respectively), tumor necrosis factor α (TNFα), intercellular adhesion molecule 1 (ICAM-1), plasminogen activator inhibitor 1 (PAI-1), or matrix-degrading proteases, through which they influence neighboring cells in a paracrine manner (reviewed in [12,13]). Due to the ongoing high rate of metabolism [2], senescent cells undergo severe morphological changes such as cell size expansion, flattening, protrusion development, chromosomal aberrations, or multinucleation [14]. Since senescent cells are known to accumulate with higher age and have also been identified in lesions of age-related diseases, including atherosclerosis [15], their involvement in the development and progression of these pathologies has been extensively discussed.

Endothelial cells lining the lumen of blood vessels are exposed to various blood-borne factors, making them susceptible to acquiring a senescent phenotype. Indeed, endothelial cells appear to be the first cell type to be affected by senescence in vivo [16]. Normally, endothelial cells are long-lived and quiescent and are only primed to proliferate during special occasions such as wound healing or pregnancy [17]. However, at sites of vascular bifurcation, carotid sinus, or aortic arch, endothelial cells are induced to proliferate by fluctuating hemodynamic forces. Endothelial cells at these sites of accelerated turnover are particularly prone to (replicative) senescence, predisposing these sites to the development of atherosclerosis [15,18]. However, quiescent endothelial cells are also prone to developing a senescent phenotype because they are exposed for very long periods of time to various noxious substances circulating in the blood, where they also encounter systemic inflammatory conditions of either an acute or chronic nature [19]. The senescent endothelial layer is characterized by substantial structural and functional changes including limited angiogenic potential [20], leaky barrier function [21], impaired vascular tone due to reduced eNOS function, and subsequent prothrombotic character [22], all of which are known to promote the progression of cardiovascular diseases.

We aimed at an unbiased integrative assessment of changes in endothelial cell programs during replicative or stress-induced senescence. Therefore, we performed a comprehensive RNA sequencing analysis of HUVECs induced to senesce by extensive cultivation or ionizing radiation and compared them with early passage HUVECs. In addition, since acute and chronic inflammation are thought to be involved in the onset and progression of senescence, we compared the transcriptomes of HUVECs treated with TNFα for short or long periods with their unstimulated counterparts and with senescent cells.

## 2. Materials and Methods

### 2.1. Isolation and Cell Culture of HUVECs

Primary human umbilical vein endothelial cells (HUVECs) were obtained in agreement with the ethical permissions GS4-EK4/562-2018 (Ethic Commission of Lower Austria) and EK 1259/20218 (Medical University of Vienna), from naval strings using a well-described collagenase digestion method [23]. HUVECs were grown on pre-gelatinized cell culture dishes in a M199-based medium supplemented with 20% FBS (Sigma-Aldrich, Saint Louis, MO, USA), 0.4% endothelial cell growth supplement with heparin (ECGS/H, PromoCell, Heidelberg, Germany), 2 mM L-glutamine, 0.1% penicillin, 0.1% streptomycin, and 0.25 μg/mL fungizone (all from Lonza, Visp, Switzerland) and cultured at 37 °C with 5% CO_2_ and 95% humidity. To overcome batch effects resulting from the individual characteristics of different donors, HUVECs of at least five single donors were pooled.

Replicative senescence of HUVECs was achieved by passaging them 35 times in a 1:3 ratio. Stress-induced premature senescent HUVECs were created by exposure to ionizing irradiation. Briefly, young, proliferating HUVECs were harvested and subjected in suspension to 10 Gy by using the cesium^137^-based irradiator IBL-437 C (kindly provided by the Department of Blood Group Serology and Transfusion Medicine, Medical University of Vienna). After irradiation, HUVECs were re-seeded on a growth area six times larger than the harvested area, to provide sufficient space for spreading caused by their expanded cell size. Irradiated HUVECs were kept in culture for ten more days before performing downstream analysis. Chronic inflammatory stimulation was conducted by exposing early passage HUVECs to 10 ng/mL TNFα (PeproTech, London, UK) for six continuous days (medium change every second day) followed by a three-day recovery period summing up to a 9 d inflammatory condition. Acute inflammation was mimicked by stimulating the cells with TNFα for 5 h only.

### 2.2. RNA Isolation and qPCR

For isolating RNA, proliferating HUVECs were seeded into gelatin-coated 6-well plates. Senescent cells (both replicative and stress-induced) were placed into 75 cm^2^ cell culture flasks as triplicates. RNA was isolated using a commercial Peqgold total RNA kit (VWR International, Vienna, Austria) according to the instructions. Determination of RNA concentration and purity (260/280 ratio > 1.8) was performed with a NanoDrop instrument (Thermo Fisher Scientific, Waltham, MA, USA). Aliquots of RNA samples were sent to the Core Facility of The Medical University of Vienna (senescent HUVECs) and the Research Center for Molecular Medicine of the Austrian Academy of Science (CeMM) (TNFα-treated HUVECs) for RNA sequencing.

To determine the relative expression values of selected genes, 450 ng of total RNA was reverse transcribed to cDNA using oligo (dT) 18 primer (Bioline, London, UK), dNTP (Thermo Fisher Scientific), RNAse Inhibitor and M-MuLV Reverse Transcriptase (both Lucigen, Middleton, WI, USA). Real-time PCR was conducted for TNFα, N-cadherin, vimentin, and fibronectin on a CFX Connect Real-Time System PCR machine using SsoAdvanced Universal SYBR Green Supermix (both Bio-Rad Laboratories, Vienna, Austria). Changes in gene expression were calculated in comparison to the control via the Pfaffl method using delta-CT values [24] or via the N0-method [25] by normalizing to glucuronidase-beta, GUSB, a housekeeping gene not affected by senescence-associated gene alterations [26], considering the respective PCR efficiencies. Primers are specified in the Appendix A.

### 2.3. Library Preparation and Sequencing

#### 2.3.1. For Senescent HUVECs

RNA-seq libraries were prepared from total RNA using the QuantSeq FWD protocol (Lexogen, Vienna, Austria) at the Core Facility Genomics department, Medical University of Vienna. PCR cycles were optimized via qPCR per the manufacturer’s instructions. Library quality and concentration were assessed using the Agilent Bioanalyzer 2100 (High Sensitivity DNA Kit) and Qubit dsDNA HS Assay Kit (Invitrogen, Thermo Fisher Scientific, Vienna, Austria), respectively. Pooled libraries were sequenced on an Illumina NextSeq500 (1 × 75 bp single-end). FASTQ files were generated with bcl2fastq (v2.19.1.403, Illumina, San Diego, USA). Adapter trimming and quality filtering were performed using cutadapt (v1.15) [27]. Samples initially yielded ~8 million reads and were downsampled to 5 million reads each for uniform depth. Reads were aligned to the GRCh38 reference genome with Gencode v29 using STAR (v2.6.1a) [28], and gene-level raw counts were obtained directly from STAR.

#### 2.3.2. For TNFα-Treated HUVECs

The amount of total RNA was quantified using a Qubit 2.0 Fluorometric Quantitation assay (Thermo Fisher Scientific, Waltham, MA, USA), and the RNA integrity number (RIN) was determined using a 2100 Bioanalyzer High-Resolution Automated Electrophoresis instrument (Agilent, Santa Clara, CA, USA). RNA-seq libraries were prepared using a NEBNext^®^ Poly(A) mRNA Magnetic Isolation Module E7490 and NEBNext^®^ Ultra™ II Directional RNA sample preparation kit E7760 (New England Biolabs, Inc., Ipswich, MA, USA). NGS library concentrations were quantified via the Qubit 2.0 assay and the size distribution was assessed using the 2100 Bioanalyzer instrument. Before sequencing, sample-specific NGS libraries were diluted and pooled in equimolar amounts. Expression profiling libraries were sequenced on a NovaSeq 6000 instrument (Illumina, San Diego, CA, USA) following a 100-base-pair, paired-end recipe. An average of 37 × 10^6^ raw sequencing reads were obtained. Reads were trimmed for quality and adapter sequences using Trim Galore (v0.6.10) [29] with a Phred score cutoff of 30 and a minimum read length of 30 bp. Reads (~36 × 10^6^ per sample) were aligned to the human genome GRCh38 release 95 primary assembly using STAR (v2.7.11b) [28], with genome indices and transcript annotations based on Ensembl release 95, and gene-level raw counts were obtained directly from STAR.

### 2.4. Bioinformatic Analysis

Differential expression analysis was conducted in RStudio Server (v.2024.12.1.563) [30] with R version 4.4.3 (28 February 2025), using the edgeR package (v.4.4.2) [31]. Prior to analysis, gene annotations were retrieved from Ensembl release 113 using the biomaRt R package (v2.62.1) [32]. Ensembl IDs from the raw gene count matrix were mapped to Entrez Gene IDs. Only genes annotated as protein coding were included in downstream analyses. Genes were considered to be differentially expressed when they showed an adjusted *p*-value < 0.05 and an absolute log2 fold change (FC) > 1. Additional analyses included protein interaction networks using NetworkAnalyst [33] with the STRING database protein interactome [34], visualization via Cytoscape 3.10.3 [35], gene overlap analysis via DeepVenn [36], and enrichment analysis with Enrichr [37,38]. ggplot2 (v3.5.1) [39] was used for data visualization.

### 2.5. Gene Set Enrichment Analysis

Gene set enrichment analysis (GSEA) [40] was used to detect coordinated changes across predefined gene sets, overcoming the limitations of gene-level testing. Genes were ranked by −log10(*p*-value) × log2FC to incorporate both direction and significance. GSEA was run using fgsea [41] from the clusterProfiler package (v4.14.6) [42], with Hallmark gene sets retrieved via msigdbr (v10.0.1) [43] from MSigDB [44].

### 2.6. Protein Isolation and Western Blot

HUVECs were treated and prepared as specified above and the protein content was extracted from 25 cm^2^ (young and TNFα-treated) and 75 cm^2^ (senescent) cell culture flasks. Briefly, cells were harvested by trypsinization, and the pellet was dissolved in ice-cold RIPA buffer (Sigma-Aldrich) containing protease inhibitor cocktail (Bimake, Houston, TX, USA). Equal amounts of proteins were loaded on 10% polyacrylamide gels for SDS-PAGE. Proteins were blotted on PVDF membranes (Carl Roth, Karlsruhe, Germany) using the wet blotting method. For detection of lamin B1 as a marker for senescence, the membranes were first blocked in 5% skimmed milk in PBS-T and incubated overnight with anti-lamin B1 rabbit primary monoclonal antibody (1:500; ab133741, Abcam, Cambridge, UK) and anti-β-actin (as reference) rabbit primary polyclonal antibody (1:1000; A2066, Sigma-Aldrich) followed by incubation with secondary donkey anti-rabbit HRP-conjugated antibody (1:5000; NA934, Amersham, UK) for an additional 2 h. Membranes were developed with HRP Western Bright Sirius substrate (Advansta, San Jose, CA, USA) and visualized using a CCD-based chemiluminescence imager (Alpha Innotech, San Leonardo, CA, USA). Protein bands were analyzed using ImageJ (version 1.53c, ref. [45]).

### 2.7. IL-6 and IL-8 ELISA

Cells were seeded into gelatin-coated 96-well plates and handled as indicated earlier. HUVECs mimicking chronic inflammation were treated intermittently every second day with 10 ng/mL TNFα for 6 days followed by 3 days of recovery in full medium, whereas acute inflammation HUVECs were exposed to TNFα for 5 h. The supernatant was centrifuged for 5 min at 3000× *g* directly after collection, to remove cellular debris, and it was stored frozen at −20 °C until it was used for the assay. For this, Ready-SET-Go! IL-8 and IL-6 ELISA kits (88-8086 and 88-7066, both Thermo Fisher Scientific) were prepared according to the manufacturer’s instructions. In brief, low-binding 96-well plates were coated with capture antibodies overnight, blocked with ELISA/ELISPOT diluent, and subsequently incubated overnight with IL-8 and IL-6 standards as well as the HUVEC supernatant samples. This was followed by incubation with detection antibodies, streptavidin–HRP linked secondary antibodies, and TMB substrate, and the reaction was ultimately stopped by adding 2N H_2_SO_4_ (Merck, Darmstadt, Germany). Absorbance was measured on a BioTek EL × 808 plate reader (Agilent, Santa Clara, CA, USA) at 450 nm and 630 nm. The measured values were normalized to the number of cells that were determined by nuclear Hoechst 33258 staining.

### 2.8. Cell Cycle Analysis by Flow Cytometry of 7AAD-Incorporation into DNA

Flow cytometry-based quantification of cells in the G0/G1-, S- and G2/M-phases of the cell cycle was carried out for senescent HUVECs and for HUVECs treated with 10 ng/mL TNFα, essentially as described in [46]. In brief, cells were harvested by trypsinization and cell pellets were resuspended in PBS and fixed by incubation in ice-cold 70% ethanol for 1 h. The ethanol was removed by centrifugation at 850× *g* for 10 min at 4 °C, followed by an additional washing step with PBS. Cells were subsequently incubated in 5 µg/mL 7-AAD (ab214663, Abcam) and 100 µg/mL RNase A for 30 min, shielded from light. Incorporation of 7AAD into DNA was quantified via flow cytometry using a CytoFLEX S cytometer and the associated CytExpert software (version 2.5, both Beckman Coulter, Indianapolis, IN, USA).

### 2.9. Wound Healing (Scratch) Assay

To assess the wound healing potential of endothelial cells under different conditions, HUVECs were seeded into 24-well plates pre-coated with 1% gelatin. They were prepared according to the 5 h or 9 d TNFα protocol, respectively, or kept in full medium in the case of control and senescent cells. All conditions were prepared in triplicates and in parallel. The wound was created by scratching the cellular monolayer with a 1000 µL pipette tip and the sample was placed immediately into an Incucyte machine (Sartorius, Göttingen, Germany) for time-lapse image acquisition. For each well, 16 images were taken every 30 min for a total observation time of 30 h. The images were analyzed with the built-in software “Confluence-AI” and the migration rate was calculated as the time needed to reach ≥95% confluence or the final confluence if 95% was not reached.

### 2.10. Immunofluorescent Staining for Ki67 as Proliferation Marker

HUVECs were seeded into gelatinized 96-well plates and treated as described above. Cells were fixed with 4% PFA (Sigma-Aldrich), permeabilized with 0.2%Triton-X 100 (Serva, Heidelberg, Germany), and blocked with 3% goat serum (Abcam) prior to adding rabbit monoclonal Ki67 antibodies (RM-9106-S1, Thermo Fisher Scientific) in a 1:500 dilution. After the overnight incubation with primary antibodies, secondary goat anti-rabbit polyclonal IgG Alexa 647 antibodies (1:500, A-21245, Thermo Fisher Scientific) were incubated for 2 h. During the final washing step, nuclear staining with 1 µg/mL Hoechst 33258 (CAY-16756-50, Cayman Chemical, Ann Arbor, MI, USA) was applied. Fluorescent images were taken on an inverted Olympus IX71 or an IX83 cellVivo live cell microscope (both Olympus, Shinjuku, Japan) with a 10× air objective, using 385 nm excitation with a 447/60 blue emission filter for DAPI/Hoechst and 660 nm with a 692/40 far-red emission filter for Ki67. Quantification of cell nuclei and Ki67-positive cells was carried out using ImageJ and respective macros.

### 2.11. Nanoparticle Analysis of Extracellular Vesicles and Secretome Analysis by MS/MS

Proliferating and replicative senescent HUVECs were seeded into 24-well plates and kept growing in full medium for five days before harvesting. For this, the supernatant was collected into microcentrifuge tubes and centrifuged first for 5 min at 300× *g* and subsequently twice at 3000× *g* for another 5 min, changing to a fresh tube each time to remove cellular debris. Next, the supernatant was filtered through a 0.2 µm PES membrane syringe filter and finally enriched using the Vivaspin 500 centrifugal concentrator with a molecular weight cut-off of 5 kDa (Sartorius) before being stored at −80 °C. Particle size and concentration were measured in 1:20 diluted samples (with full growth medium as reference control) on a tunable resistance pulse sensing (TRPS) EXOID system (Izon Science, Lyon, France). Briefly, by applying different pressure forces (1300 Pa, 1500 Pa and 1900 Pa) and a constant voltage of 604 mV, the resistive pulse of the specimen on a 150 nm nanopore was measured. The crude extracellular vesicle content was calculated by matching the sample measurements with measurements of calibration beads (TKP200), using Izon Data Suite software 1.0 (both Izon Science). For the analysis of secretomes from young and old HUVECs (comprising both extracellular vesicles and secreted protein), confluent plates were stimulated for 45 min with PMA (10 ng/mL PMA) to promote secretion and membrane fusion (three biological replicates for each condition). Supernatants were cleared from cell debris by differential centrifugation (500× *g*, 5 min and 3000× *g*, 5 min) and subjected to TCA precipitation. Protein pellets were solubilized in urea, digested, and analyzed using a Q-Exactive HF (Thermo Fisher Scientific) operated in DDA mode, and analyzed with Proteome Discoverer. Comparative statistical analysis was conducted on the basis of normalized spectral counts in Perseus by means of one-way ANOVA with Tukey’s post hoc HSD test to correct for multiple testing. A q-value < 0.05 was considered significant. Proteins identified exclusively in young or old secretomes were summarized in Venn diagrams. Enrichr (https://maayanlab.cloud/Enrichr/ accessed on 17 March 2025) was utilized for qualitative Reactome pathway enrichment analysis.

### 2.12. Statistics

For group comparisons, one-way ANOVA with Bonferroni correction was performed using GraphPad Prism 10 (Version 10.4.1 for Windows, GraphPad Software, Boston, MA, USA). If not indicated differently in the figure legends, graphs show mean values with SEM, and significances are indicated by asterisks as follows: ns > 0.05, * *p* ≤ 0.05, ** *p* ≤ 0.01, *** *p* ≤ 0.001, and **** *p* ≤ 0.0001.

## 3. Results

### 3.1. Effects of Replicative and Stress-Induced Senescence on RNA Expression

First, we wanted to obtain an unbiased overview of gene expression changes in endothelial cells during replicative senescence and compare them with the effects of premature senescence induced by gamma irradiation. Endothelial cells reflecting an aged phenotype were obtained by exhaustive replication over 35 cell passages. Stress-induced senescence was achieved by irradiating young proliferating HUVECs with 10 Gy. RNA was extracted from triplicate treatments using young proliferating endothelial cells as controls and sequenced at the university core facility. Differential gene expression analysis and subsequent bioinformatics were performed as described in the Methods section. Changes in signaling pathways and molecular signatures were determined by gene set enrichment analysis (GSEA), which has the advantage of not requiring a threshold of up- or downregulation to detect changes in whole gene sets [40]. This analysis revealed strong similarities between stress-induced and age-related senescence (Figure 1). Epithelial (or in this case endothelial)–mesenchymal transition (EMT) and a large panel of inflammatory pathways were upregulated, while cell cycle progression and DNA repair were downregulated.

### 3.2. Effects of Acute and Chronic Endothelial Inflammation on RNA Expression

Given the strong inflammatory phenotype of endothelial cells after both stress-induced and replicative senescence, we wanted to compare these specific RNA expression profiles with those of endothelial cells after short- or long-term inflammatory activation. HUVECs were treated with TNFα for 5 h to mimic acute inflammation or with TNFα for 6 days followed by 3 days in normal medium to mimic long-term inflammation after a total of 9 days. Comparison of gene expression with that of untreated young endothelial cells revealed the expected upregulation of TNFα- or interferon-related pathways and downregulation of E2F- or MYC-related proliferation pathways in both cases (Figure 2). However, EMT was more pronounced after long-term inflammation. We then directly compared the effects of 9 d TNFα treatment with those of 5 h TNFα treatment, to detect the specific differences between chronic and acute inflammation. Among the hallmark gene collection of the Molecular Signature database, only EMT genes remained as upregulated, while TNFα and interferon-related inflammatory pathways were seen as significantly downregulated after long-term TNFα treatment, indicating that the inflammatory response was significantly reduced, leaving endothelial–mesenchymal transition as one of the major consequences of chronic inflammation.

### 3.3. Overlaps and Differences Between Senescence, Acute and Chronic Inflammation

To elucidate the similarities and differences between senescent and short- and long-term TNFα-treated endothelial cells, we computed proportional Venn diagrams and queried the various overlaps using the Enrichr platform. This revealed that all three conditions overlapped (Figure 3). However, inflammatory pathways were seen only in the overlap between senescence and short-term TNFα treatment, whereas chronic inflammatory activation and senescence showed similarities in EMT, TGFβ signaling, and cell junctions.

Further analysis and visualization of differences and similarities between senescence and short- and long-term TNFα treatment was performed using the NetworkAnalyst [33] platform. Results were compared with the STRING database interactome and visualized using Cytoscape [35], which confirmed the Venn graph and pathway analysis (Appendix A).

### 3.4. Experimental Validation of the Antiproliferative and Pro-Inflammatory Effects

The results of our unbiased transcriptomic analyses prompted us to experimentally investigate some of the predicted effects of senescence and short- and long-term endothelial inflammation. To analyze proliferative potential, we stained for the cell cycle marker Ki67, which is not expressed in cells in the resting G0 phase of the cell cycle. Both types of TNFα treatment (5 h and 9 d) resulted in a similar reduction in Ki67 expression expressed as a percentage of total cells (Figure 4A). An even greater reduction in the proliferation marker was observed in irradiation-induced senescence and in replicative senescence. Representative microscopic images of the staining are shown in Appendix A. The anti-proliferative effect of TNFα was also confirmed by cell cycle analysis using incorporation of 7AAD into DNA and quantification via flow cytometry (Figure 5). As markers of inflammation, we quantified the mRNA expression of TNFα via qPCR (Figure 4B), and the protein expression and secretion of the cytokines IL-6 and IL-8 into the supernatants of the cells via ELISA. A very significant increase (when normalized to cell number) was observed in old senescent HUVECs (Figure 4C,D), confirming a pro-inflammatory effect of senescence. IL-8 secretion was elevated after TNFα treatment for 5 h, but this increase was no longer visible at 9 d after TNFα treatment, consistent with the observation from RNA sequencing results that long-term inflammation leads to a downregulation of the inflammatory response. For IL-6, only a trend towards elevation was observed after both types of TNFα treatment, which was not statistically significant. A possible explanation is that the 5 h treatment was too short to detect a measurable upregulation at the protein level and that the putative increase in gene expression and secretion did not last long enough to be measured after 9 d, especially considering that the treatment medium was replaced with fresh medium after 6 d, followed by 3 d without TNFα. Thus, cytokines released during the first 6 days were removed and the remaining 3 days may not have been sufficient to induce significant IL-6 or IL-8 secretion in the case of a rather transient inflammatory response. Furthermore, the sensitivity of the ELISA measurement may have been too low to detect mild upregulation, as the secreted cytokines were diluted in the supernatants.

### 3.5. Markers of Mesenchymal Transition Are Upregulated by TNFα Treatment or Senescence

Analysis of gene expression changes via RNA sequencing suggested a significant stimulation of epithelial or endothelial–mesenchymal transition. Well-accepted markers of this biological process, in which epithelial or endothelial cells dedifferentiate into a mesenchymal state, are the genes N-cadherin, vimentin, and fibronectin [47]. Therefore, we measured the expression of these genes by sensitive quantitative PCR after reverse transcription. The highest upregulation of all three markers was observed in cells with irradiation-induced senescence (Figure 6). All of them were also significantly upregulated in old endothelial cells undergoing replicative senescence. N-cadherin and fibronectin showed similarly increased expression in 9 d TNFα-treated cells, whereas no significant change was observed for vimentin under these conditions. None of the EMT marker genes showed an increase after 5 h TNFα treatment, in agreement with bioinformatics results indicating that EMT is predominantly seen after long-term inflammation as well as senescence.

### 3.6. The Senescence Marker Lamin Is Upregulated by Long-Term TNFα Treatment

A well-established marker of cellular senescence is the protein lamin B1 [48], which belongs to the group of intermediate filament proteins that stabilize the inner side of the nuclear membrane, and which play a role in chromatin stability and telomer function. To investigate the effects of our treatment conditions on this marker, we performed Western blot analysis of extracts from endothelial cells treated with TNFα for 5 h or 9 d and compared them with extracts from cells undergoing replicative or stress-induced senescence. As expected, the protein was almost completely lost in both types of senescence (Figure 7). Interestingly, it was upregulated after 5 h of TNFα treatment, whereas it was significantly downregulated after 9 d of TNFα treatment. This is consistent with the notion that short-term inflammation does not act as a driver of cellular senescence, while chronic inflammation already has a senescence-promoting effect.

### 3.7. Wound Healing Capacity of Senescent or TNFα-Treated Endothelial Cells

Since some cell migration-related pathways and molecular signatures were seen in the transcriptomic analysis, we performed scratch assays and long-term microscopic evaluation. Monolayers of endothelial cells treated with TNFα for 5 h or 9 d or in replicative senescence were wounded by scratching with a pipette tip, followed by microscopic imaging for 30 h. Wound closure was quantitatively assessed using the Incucyte instrument software. Short-term TNFα treatment resulted in an apparent reduction in migration rate, which was even more pronounced and statistically significant after long-term TNFα treatment (Figure 8 and Appendix A). Cells in replicative senescence could hardly close the wound during the observation period and moved more randomly in all directions, whereas cells with the other treatments showed a more directed movement towards the gap.

Since senescence is associated with a secretory phenotype, we performed an exploratory proteomic analysis of secretomes obtained from old (replicatively senescent) and young HUVECs. In this analysis, we identified a total of 427 proteins with statistical significance, of which 197 were present only in the secretomes of young HUVECs and 35 only in the secretomes of senescent cells. In total, 195 proteins were found in both conditions (Appendix A). Quantitative comparison of the commonly identified proteins based on spectral counts revealed 69 proteins significantly higher in the secretome of old and 9 proteins significantly higher in the secretome of young HUVECs (*p* < 0.05, with a log2-fold change cut-off > |0.5|), as shown in Appendix A. Notably, proteins that were higher or found only in old HUVECs had functions in hemostasis (101 proteins, adj. *p*-value 4.2 × 10^−14^ odds ratio 10.31, combined score 381). Conversely, proteins that were higher or found only in young HUVECs had functions enriched in axon guidance (198 proteins, adj. *p*-value 9.9 × 10^−20^, odds ratio 9.43, combined score 476.4).

In general, secretomes contain both secreted proteins (from secretory granules) and proteins within extracellular vesicles that are released into the supernatant. Since RNA sequencing also suggested potential effects on the formation of extracellular vesicles, we also performed a scouting experiment using a nanoparticle analyzer to determine potential differences in the formation of these vesicles. We compared only old (replicatively senescent) HUVECs with young endothelial cells. The average size of the vesicles was slightly below 100 nm regardless of the cell state. However, the number of vesicles released per cell was significantly higher in senescent cells. Together with the known diameter distribution, a significantly higher volume of the sum of these vesicles was calculated for cells in replicative senescence (Appendix A).

## 4. Discussion

As living organisms age, they are exposed to a variety of environmental and internal stressors. In this study, we aimed to conduct an integrative study to assess similarities and differences between two major types of cellular senescence and acute and chronic inflammation, which are important triggers or cofactors of cellular aging. To this end, we first compared the transcriptome of proliferation-competent young HUVECs with that of senescent endothelial cells induced into this state by either exhaustive replication or ionizing radiation stress. Both senescent states showed upregulation of genes involved in mesenchymal transition, but also inflammatory molecular signatures related to interferon and TNFα-induced NF-κB signaling (Figure 1). There were few differences between the two types of senescence, with angiogenesis, myogenesis, and coagulation upregulated in stress-induced senescence and hypoxia, and glycolysis specifically upregulated in replicative senescence. The predominance of inflammation-associated molecular signatures in both types of senescence suggested a comparison with pathways altered by acute or chronic inflammation. Therefore, we first analyzed the effects of short-term and long-term TNFα treatment compared with untreated controls (Figure 2). As in senescent cells, we observed very similar changes in molecular signatures, with the expected inflammatory pathways being enriched and MYC-related cell proliferation pathways being reduced. However, the magnitude of the molecular changes appeared to be greater in the acute inflammatory state. When we compared the long-term TNFα treatment with the acute inflammatory state, we found that mesenchymal transition was more predominant in chronic inflammation, whereas the classical inflammatory signatures and pathways induced by UV were already downregulated after long-term TNFα treatment. This suggests that acute defense mechanisms against stressors are reduced upon prolonged stimulation and replaced by phenotypic changes and alterations in cellular differentiation states. Separate analysis of the effects of senescence and inflammation revealed that they induced similar responses with overlapping signatures. We analyzed this aspect by visualizing the transcriptomic changes and their overlap by using Venn graph tools and querying the different intersections as well as unique gene sets via pathway enrichment methods (Figure 3). This revealed that the similarity between short-term TNFα treatment and senescence mainly included the classical TNFα and interferon-related inflammatory signatures, while the overlap between senescence and long-term TNFα treatment was related to mesenchymal transition, TGFβ signaling, and myogenesis. This may suggest that inflammation drives senescence pathways under both acute and chronic conditions, albeit at different mechanistic levels that may act cooperatively. However, these findings must be interpreted with caution, as the transcriptomic profile of HUVECs changed after the cells were explanted from the intact umbilical cord and cultured in vitro, due to the altered microenvironment with a lack of blood flow and lack of crosstalk with other cells [49]. Nevertheless, our results are consistent with a meta-analysis of gene expression changes associated with aging that identified inflammatory pathways as an important common denominator [50]. Furthermore, we were able to verify our bioinformatic analyses with experimental results that supported the predicted changes in cell proliferation and expression of inflammatory cytokines (Figure 4 and Figure 5) or genes involved in mesenchymal transition (Figure 6). Similarly, we were able to show that the marker lamin B1, which is lost in senescence, is reduced after long-term TNFα treatment (Figure 7), in agreement with some other reports [51]. Interestingly, however, lamin B1 was significantly elevated after short-term TNFα treatment, suggesting that chronic inflammation is required to trigger its reduction. Finally, our studies showed that inflammatory activation of endothelial cells has similar functional consequences in the context of wound healing—a combination of proliferation and migration of cells (Figure 8). In particular, long-term TNFα treatment and cellular senescence had an inhibitory effect on wound healing potential. Since both chronic inflammation and cellular senescence increase with age, this explains the age-related decrease in regenerative potential.

## 5. Conclusions

Our study shows that transcriptomic changes in endothelial cells after radiation stress-induced senescence are very similar to those induced by replicative senescence. Both situations also prominently induce inflammatory pathways. Induction of acute or chronic inflammation with short- or long-term TNFα treatment revealed similarities but also differences, with a reduction of the classical inflammatory signature under chronic inflammatory conditions while a dominant signature of mesenchymal transition remained. Molecular signatures of inflammation overlapped significantly with those of senescence. Experimental validation of the transcriptomic analysis confirmed the anti-proliferative effect of TNFα treatment and the pro-inflammatory effect of senescence. Furthermore, we demonstrated that markers of mesenchymal transition and senescence were upregulated by long-term TNFα treatment and that both inflammatory cell activation and senescence reduced the wound healing potential of endothelial cells.

## Figures and Tables

**Figure 1 cells-14-00806-f001:**
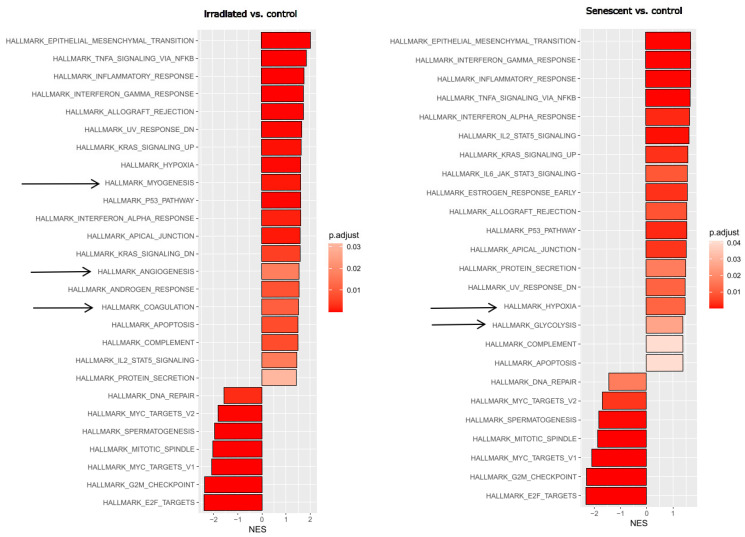
Gene set enrichment analysis (GSEA) after stress-induced or replicative senescence of endothelial cells. Left: Young HUVECs irradiated with 10 Gy (at a stress-induced senescent state) were compared to non-irradiated control cells. Right: HUVECs at replicative senescence (after 35 passages) compared with young endothelial control cells. Significant hallmark gene sets from the Molecular Signature database are shown with the normalized enrichment score (NES), with the adjusted *p*-value shown in gradients of red as indicated in the legend. Arrows indicate differences between the responses. Three replicates were analyzed for each experimental condition.

**Figure 2 cells-14-00806-f002:**
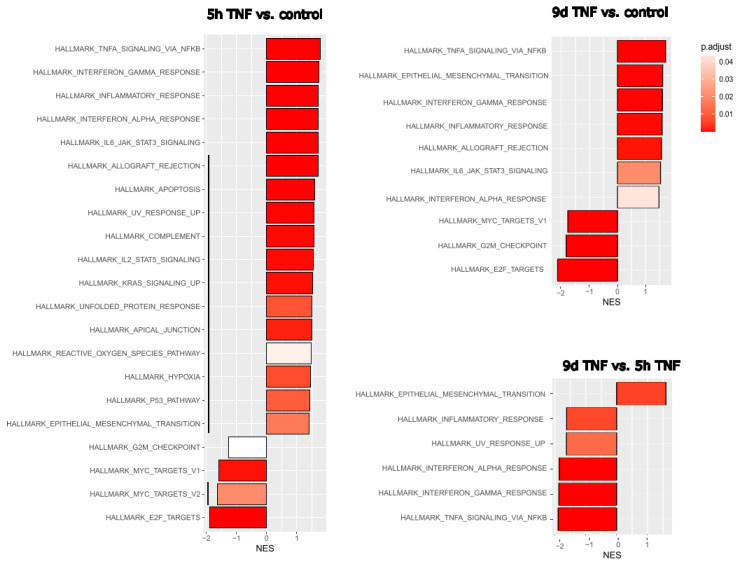
Gene set enrichment analysis (GSEA) after short- or long-term TNFα treatment of endothelial cells. Left: Hallmark gene set changes of HUVECs treated for 5 h with TNFα compared with untreated controls. Upper right: Hallmark gene set changes of HUVECs after long-term TNFα treatment versus untreated controls. Normalized enrichment scores (NES) are shown with the adjusted *p*-value in gradients of red as indicated in the legend. Black lines indicate differences in the molecular signatures altered by 5 h and 9 d TNFα treatment. Lower right: Direct comparison of 9 d versus 5 h TNFα treatment for a more sensitive assessment of their differences. Three replicates were analyzed for each experimental condition. Black vertical lines indicate differences in enriched signatures.

**Figure 3 cells-14-00806-f003:**
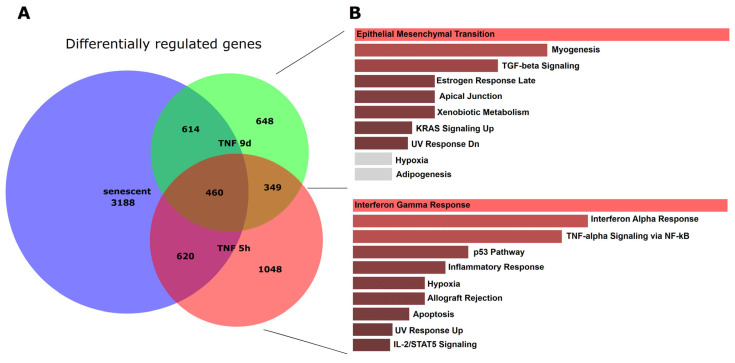
Proportional Venn graph showing gene expression changes, their overlaps, and enriched pathways for senescent and short- and long-term TNFα-treated endothelial cells. (**A**) Proportional Venn graph for significantly altered genes (FDR < 0.05; log-fold change above 0.5 or below −0.5) using DeepVenn [36]. (**B**) Pathways enriched for the overlap between senescence and 9 d TNFα treatment, and those for the overlap between senescence and 5 h TNFα treatment, were determined using the Enrichr web platform [38].

**Figure 4 cells-14-00806-f004:**
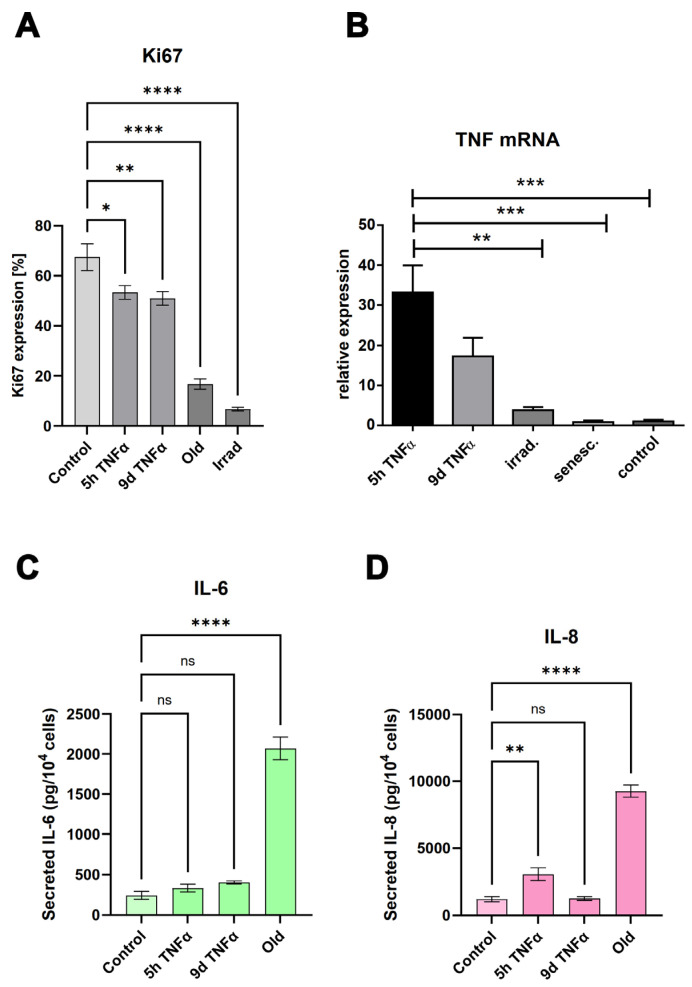
(**A**) Quantification of the proliferation marker Ki67 in young endothelial cells treated with TNFα for 5 h or 9 d, compared with cells in replicative senescence (old) and stress-induced senescence (irradiated). Mean values +/− SEM (N = 8). (**B**) Quantification of TNFα mRNA expression by quantitative RT-PCR (relative expression compared with control, mean +/− SEM, N = 3). (**C**,**D**) Quantification of IL-6 secretion (**C**) or IL-8 secretion (**D**) in cells under the conditions described in (**A**). Bars represent mean values +/− SEM (N = 4). Asterisks represent significance as specified in the Statistics Section 2.12.

**Figure 5 cells-14-00806-f005:**
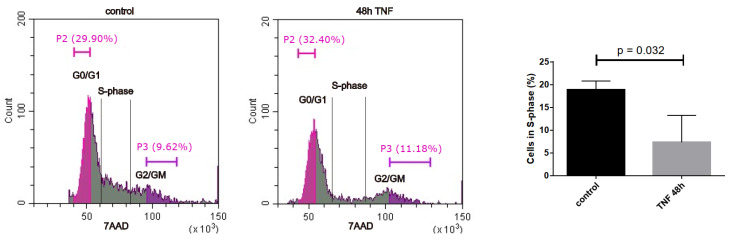
Cell cycle analysis via flow cytometry with 7AAD incorporation into DNA. Representative flow cytometry profiles are shown for control and 48 h TNFα. Half of the G0/G1 and the G2/M peak was quantified as indicated to determine the percentage of cells in S-phase. Right panel: Quantification of cells in S-phase (mean +/− SEM, N = 3).

**Figure 6 cells-14-00806-f006:**
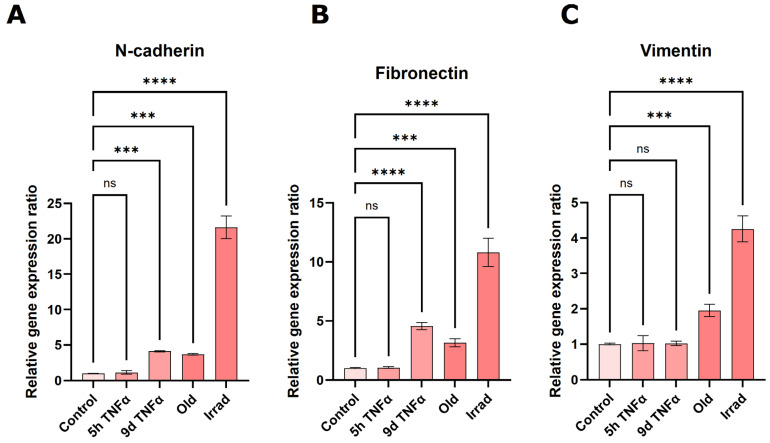
qPCR-based quantification of the expression of the EMT-marker genes N-cadherin (**A**), fibronectin (**B**), and vimentin (**C**) in young HUVECs treated with TNFα for 5 h or 9 d, HUVECs in replicative senescence (old) or young HUVECs rendered senescent by gamma irradiation (irrad). Mean values +/− SEM (N = 3), asterisks indicate significance as described under Section 2.12.

**Figure 7 cells-14-00806-f007:**
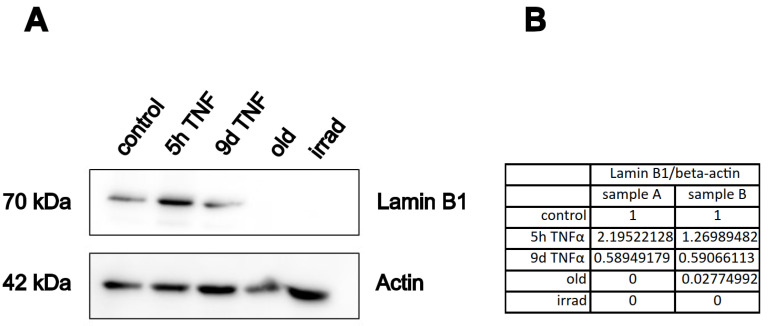
Western blot analysis of the senescence marker lamin B1. HUVECs were treated with TNFα or rendered senescent as indicated (Old: replicative senescence, Irrad; irradiation-induced senescence), followed by SDS-PAGE and immuno-blotting with antibodies against lamin B, or actin as loading control. (**A**) Representative Western blot. (**B**) The lamin B1 band was normalized to the actin band for quantification of two independent experiments. The table shows the relative lamin B1 expression normalized to beta-actin for the two samples.

**Figure 8 cells-14-00806-f008:**
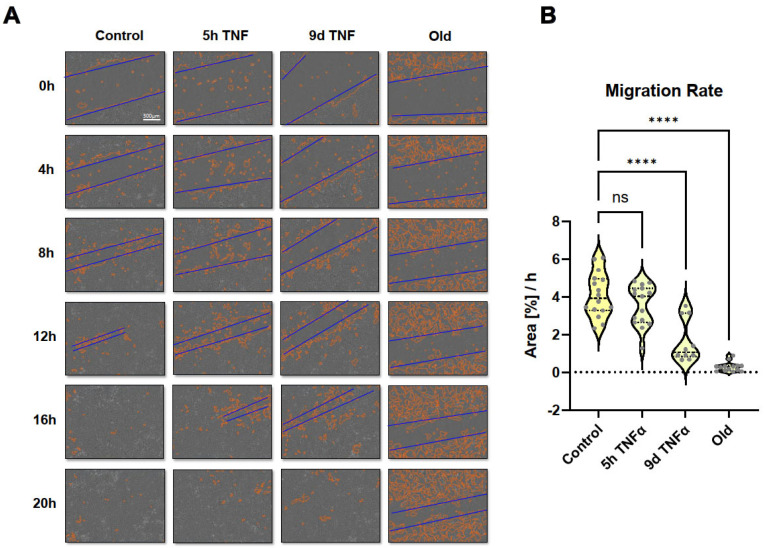
Wound healing (scratch) assay using long-term observation with an Incucyte system. HUVECs were left untreated (control) or were treated with TNFα for 5 h or 9 d and were compared with cells in replicative senescence (old). (**A**) Representative microscopy images showing automatically recorded cell front borders in red and approximate dimensions of the gaps in blue. (**B**) Quantification of the recorded migration rates as assessed by area closure in %/h. Violin plot distribution (N = 16), asterisks indicate significance as described under Section 2.12.

## Data Availability

The data presented in this study are available upon request from the corresponding author. The RNA sequencing datasets have been deposited to the NCBI’s Gene Expression Omnibus GEO database and are accessible through GEO Series accession numbers GSE296602 (https://www.ncbi.nlm.nih.gov/geo/query/acc.cgi?acc=GSE296602) and GSE294733 (https://www.ncbi.nlm.nih.gov/geo/query/acc.cgi?acc=GSE294733).

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
