# Peer review of "Short- and Long-Term Endothelial Inflammation Have Distinct Effects and Overlap with Signatures of Cellular Senescence"

_cells, 2025, doi:10.3390/cells14110806_

Round 1

Reviewer 1 Report

Comments and Suggestions for Authors

The manuscript by Barbora Belakova et al investigated the gene expression profiles in HUVEC under replicative- radiation stress- or cytokine-induced senescence conditions. The results have also been validated at protein expression levels. This is a well-designed study with a goal to compare and uncover the differences of senescent endothelial cells. The results will provide important information to the field. 

Author Response

We thank the reviewer for this positive evaluation of our manuscript.

Reviewer 2 Report

Comments and Suggestions for Authors

The study by Belakova et al. investigates molecular and functional differences/similarities in cultured human umbilical vein endothelial cells undergoing senescence (replicative and radiation-induced) or inflammation (acute and chronic). The study is clearly written, and the topic is undoubtedly of great biomedical interest. Some aspects of the manuscript could be significantly improved.

  1. The authors should provide a q-PCR analysis of TNFα gene expression and regulation in both types of senescence (replicative and radiation-induced) compared to untreated young endothelial cells, and cells treated with TNFα. TNFα is a canonical component of the SASP, and it is necessary to determine whether the observed similarities between both conditions (senescence/inflammation) are explained by similar mechanisms.
  2. A complementary PI flow cytometry analysis would be useful to complement the cell proliferation and death data to confirm the senescence status.
  3. The manuscript is excessively long, which makes it difficult to follow the descriptions (the methodological descriptions are very detailed and many aspects could be eliminated by making appropriate references. i.e supplementary tables summarizing the oligos employed would be of help to reduce text).
  4. Both in the description of qPCR experiments in methods section and in the figure showing the results, the authors must specify that the quantification method is Relative. Given the employment of Pfaffl analysis, it would be more correct to indicate "Relative Gene Expression Ratio" rather than “Fold Change” often employed for Livak analysis.

Author Response

The study by Belakova et al. investigates molecular and functional differences/similarities in cultured human umbilical vein endothelial cells undergoing senescence (replicative and radiation-induced) or inflammation (acute and chronic). The study is clearly written, and the topic is undoubtedly of great biomedical interest. Some aspects of the manuscript could be significantly improved.

  1. The authors should provide a q-PCR analysis of TNFα gene expression and regulation in both types of senescence (replicative and radiation-induced) compared to untreated young endothelial cells, and cells treated with TNFα. TNFα is a canonical component of the SASP, and it is necessary to determine whether the observed similarities between both conditions (senescence/inflammation) are explained by similar mechanisms.

    We added a qPCR analysis of TNFα-mRNA expression, which is now included in the revised version of Figure 4 (panel B).
  2. A complementary PI flow cytometry analysis would be useful to complement the cell proliferation and death data to confirm the senescence status.

    We performed a flow cytometry cell cycle analysis using 7AAD (as less toxic alternative to PI), which is now included in the new Figure 5. Due to very low numbers of cells in the senescent sample, we could not generate sufficiently robust data for this condition. Nevertheless, the anti-proliferative effect of TNFα could be confirmed by a reduction of the percentage of cells in the S-phase of the cell cycle.
  3. The manuscript is excessively long, which makes it difficult to follow the descriptions (the methodological descriptions are very detailed and many aspects could be eliminated by making appropriate references. i.e supplementary tables summarizing the oligos employed would be of help to reduce text).

    We shortened the bioinformatics part of the Methods section significantly, and also summarized the PCR-primer information in a table in the Supplementary Information.
  4. Both in the description of qPCR experiments in methods section and in the figure showing the results, the authors must specify that the quantification method is Relative. Given the employment of Pfaffl analysis, it would be more correct to indicate "Relative Gene Expression Ratio" rather than “Fold Change” often employed for Livak analysis.

    We changed the text accordingly (in the main text, as well as in figure legends).

Reviewer 3 Report

Comments and Suggestions for Authors

The paper by Belakova and colleagues reports the gene expression changes in human endothelial cells (HUVECs) subjected to replicative- or radiation-stress-induced senescence, and compared these profiles with those of cells under acute or chronic TNFα-mediated inflammation. The authors found that both senescence types induced upregulation of genes associated with epithelial- (or endothelial) mesenchymal transition (EMT) and inflammatory pathways, indicating a common molecular signature. In addition, chronic inflammation led to a pronounced EMT signature, while acute inflammation activated classical inflammatory responses. From a functional point of view, senescent and chronically inflamed cells showed reduced proliferation and increased secretion of pro-inflammatory cytokines (IL-6 and IL-8). An impaired wound healing capacity was revealed in senescent and chronically inflamed cells, thus explaining the age-related decrease in tissue regenerative potential.

Comments: the topic covered by the paper is novel and uptodate. The experimental design and technologies used for genomic signature evaluation are appropriate. Results sound and are fitting with the physiopathology of vascular endohelium in ageing and inflammatory related disorders.

Minor comments: Please complete the list of abbreviations.

Author Response

We thank the reviewer for this positive assessment of our manuscript. The list of abbreviations has been completed in the revised manuscript.
